Linking pangenomes and metagenomes: the Prochlorococcus metapangenome

Delmont Tom O. tomodelmont@gmail.com 1
Eren A. Murat a.murat.eren@gmail.com meren@uchicago.edu 1 2
1 Department of Medicine, University of Chicago , Chicago , IL , United States of America
2 Josephine Bay Paul Center, Marine Biological Laboratory , Woods Hole , MA , United States of America
Bajic Vladimir
Electronic publication date: 2018 Jan 25
Publication date: 2018
Volume: 6
Electronic Location ID: e4320
Received 2017 Oct 13; Accepted 2018 Jan 13
Copyright: ©2018 Delmont and Eren
Copyright year: 2018
Copyright holder: Delmont and Eren
License: This is an open access article distributed under the terms of the Creative Commons Attribution License, which permits unrestricted use, distribution, reproduction and adaptation in any medium and for any purpose provided that it is properly attributed. For attribution, the original author(s), title, publication source (PeerJ) and either DOI or URL of the article must be cited.
License URL: https://creativecommons.org/licenses/by/4.0/

Keywords: Comparative genomics, Metagenomics, Microbial ecology, Metapangenomics, anvi’o, Hypervariable genomic islands, Sugar metabolism, Pangenomics, TARA Oceans

Funding: University of Chicago This work was supported by the Frank R. Lillie Research Innovation Award, and startup funds from the University of Chicago. The funders had no role in study design, data collection and analysis, decision to publish, or preparation of the manuscript.

==============================
Pangenomes offer detailed characterizations of core and accessory genes found in a set of closely related microbial genomes, generally by clustering genes based on sequence homology. In comparison, metagenomes facilitate highly resolved investigations of the relative distribution of microbial genomes and individual genes across environments through read recruitment analyses. Combining these complementary approaches can yield unique insights into the functional basis of microbial niche partitioning and fitness, however, advanced software solutions are lacking. Here we present an integrated analysis and visualization strategy that provides an interactive and reproducible framework to generate pangenomes and to study them in conjunction with metagenomes. To investigate its utility, we applied this strategy to a Prochlorococcus pangenome in the context of a large-scale marine metagenomic survey. The resulting Prochlorococcus metapangenome revealed remarkable differential abundance patterns between very closely related isolates that belonged to the same phylogenetic cluster and that differed by only a small number of gene clusters in the pangenome. While the relationships between these genomes based on gene clusters correlated with their environmental distribution patterns, phylogenetic analyses using marker genes or concatenated single-copy core genes did not recapitulate these patterns. The metapangenome also revealed a small set of core genes that mostly occurred in hypervariable genomic islands of the Prochlorococcus populations, which systematically lacked read recruitment from surface ocean metagenomes. Notably, these core gene clusters were all linked to sugar metabolism, suggesting potential benefits to Prochlorococcus from a high sequence diversity of sugar metabolism genes. The rapidly growing number of microbial genomes and increasing availability of environmental metagenomes provide new opportunities to investigate the functioning and the ecology of microbial populations, and metapangenomes can provide unique insights for any taxon and biome for which genomic and sufficiently deep metagenomic data are available.

Introduction

During the last two decades, the genomic content of more than 100,000 microbial isolates has been characterized and used to study the gene pool, adaptation capabilities, and evolution of microorganisms (Smith et al., 1997; Alm et al., 1999; Makarova et al., 2006; Kumar et al., 2011; Fernández-Gómez et al., 2013). Cultivation-based approaches have paved the way for the emergence of powerful strategies to identify core and accessory genes shared between closely related genomes through pangenomics (Read et al., 2003; Tettelin et al., 2005; Zhu et al., 2015). Genomic comparisons of isolates can shed light on the biogeographic partitioning of variable genes within microbial lineages based on isolation source (Reno et al., 2009; Porter et al., 2016). Yet de novo investigations of the role of genomic traits in the adaptation of microorganisms to the environment remain difficult as cultivation alone does not offer insights into the abundance or distribution patterns of isolated populations.

Shotgun metagenomics, the sequencing of DNA directly extracted from the environment (Handelsman et al., 1998), allows the study of microbial communities without the need for cultivation. As of today, metagenomic data originating from a wide range of ecosystems make up a large fraction of the sequences stored in public databases (Qin et al., 2010; Bork et al., 2015). Researchers have used metagenomics to discover new bioactive molecules (Lorenz & Eck, 2005; Thies et al., 2016), investigate the functional potential of ecosystems (Tringe et al., 2005; Al-Amoudi et al., 2016), and access the genomic context of uncultivated microorganisms (Tyson et al., 2004; Haroon et al., 2016; Delmont et al., 2017). Metagenomic data also provide a means to quantify the abundance and relative distribution of genomes in environmental samples through read recruitment (Tyson et al., 2004; Dutilh et al., 2014; Eren et al., 2015). Although the environmental signal resulting from such analyses provides insights into the ecological niche of individual populations (Sharon et al., 2013; Bendall et al., 2016; Anderson et al., 2017; Quince et al., 2017), this approach alone does not reveal to what extent genes that may be linked to the ecology and fitness of microbes are conserved within a phylogenetic clade.

Recently, pangenomic approaches have been used to characterize the gene content of microbial populations in environmental samples through metagenomic read recruitment (Delmont & Eren, 2016; Scholz et al., 2016; Nayfach et al., 2016). Combining well-established practices from pangenomics (identifying gene clusters and inferring relationships between genomes based on shared genes), with the emerging opportunities from metagenomics (the ability to track populations precisely across environments through genome-wide read recruitment) could provide a framework to investigate the ecological role of gene clusters that may be linked to the niche partitioning and fitness of microbial populations. To explore the potential of this concept, we developed a novel workflow within an existing open-source software platform (Eren et al., 2015), and characterized the metapangenome of Prochlorococcus isolates and single-cell genomes on a large scale.

Prochlorococcus is an extensively studied photosynthetic bacterial taxon abundant in the euphotic zone of low latitude marine systems (Chisholm et al., 1988; Olson et al., 1990; Rusch et al., 2010), which fixes a substantial amount of carbon from the atmosphere (Flombaum et al., 2013). Cultivation efforts targeting Prochlorococcus resulted in the recovery of genomes that represent members from five major phylogenetic clades divided into groups that are adapted to high-light (sub-clades HL-I and HL-II) or low-light (sub-clades LL-I, LL-II, LL-III, and LL-IV) (Biller et al., 2014a). Environmental surveys and culture experiments revealed the ecological niche and temporal dynamics of HL and LL Prochlorococcus ecotypes in the oceans, as well as correlations between the genomic traits of isolates and their response to environmental variables (West et al., 2001; Rocap et al., 2003; Malmstrom et al., 2010). A previous study by Coleman & Chisholm (2010) used a pangenome of 12 Prochlorococcus isolates to discuss the differential occurrence in Prochlorococcus populations between two sampling stations after identifying core versus accessory genes and observing that only a few genes differed significantly in abundance between the sites. In addition, Kent et al. (2016) showed a strong association between the Prochlorococcus accessory gene functions and the community composition of this lineage on a large scale using metagenomes from the Global Ocean Sampling expedition. Yet to the best of our knowledge, pangenomes have never been linked to metagenomes at an appropriate resolution to monitor the distribution of individual gene clusters. Monitoring individual gene clusters is essential to scrutinize their prevalence across multiple microbial genomes, and infer associations regarding their potential role in fitness and niche partitioning of microbial populations to which they belong.

Here we investigated the gene clusters we identified in 31 Prochlorococcus isolates in conjunction with their occurrence in the surface of marine systems using 30.9 billion metagenomic reads from the TARA Oceans Project (Sunagawa et al., 2015). Our investigation revealed that closely related Prochlorococcus populations sharing the same high-light niche (i.e., near the surface) exhibit considerable differences in their relative abundance that could be explained by a small number of differentially occurring gene clusters. Finally, we extended our analysis of 31 isolates with 74 single-amplified genomes (SAGs) and revealed intriguing patterns within Prochlorococcus hypervariable genomic islands by quantifying the link between individual gene clusters and the environment

Materials and Methods

The URL http://merenlab.org/data/2018_Delmont_and_Eren_Metapangenomics/ contains a reproducible workflow that extends the descriptions and parameters of programs used in our study to (1) compute the Prochlorococcus pangenome using 31 isolate genomes, (2) profile reads isolate genomes recruited from metagenomes, and (3) generate a metapangenome for Prochlorococcus.

Genomes and metagenomes

We acquired 31 isolate genomes and 74 SAGs (minimum length >1 Mbp) of Prochlorococcus from the National Center for Biotechnology Information (NCBI), and downloaded 93 TARA Oceans metagenomes from the European bioinformatics institute (EBI) repositories. Table S1 reports accession numbers and other information for each isolate genome, SAG and metagenome.

Data preparation, quality filtering, and read recruitment

We removed the low-quality reads from the TARA Oceans dataset using ‘iu-filter-quality-minoche’, which is a program in illumina-utils v1.4.1 (Eren et al., 2013) (available from https://github.com/merenlab/illumina-utils), which implements the noise filtering parameters described by Minoche, Dohm & Himmelbauer (2011). After simplifying the header lines of 31 FASTA files for Prochlorococcus isolate genomes using the anvi’o script ‘reformat-fasta’, we concatenated all FASTA files into a single file, and used Bowtie2 (Langmead & Salzberg, 2012) with default parameters and the additional ‘--no-unal’ flag to recruit quality-filtered short metagenomic reads on to Prochlorococcus isolate genomes (‘read recruitment’ is an analogous term to ‘mapping’, or ‘short read alignment’). We used samtools (Li et al., 2009) to convert resulting SAM files into sorted and indexed BAM files.

Phylogenomic analysis

We used Phylosift v1.0.1 (Darling et al., 2014) with default parameters to quantify evolutionary distances between genomes. Briefly, Phylosift (1) identifies a set of 37 marker gene families in each genome, (2) concatenates the alignment of each marker gene family across genomes, and (3) computes a phylogenomic tree from the concatenated alignment using FastTree 2.1 (Price, Dehal & Arkin, 2010). We finalized the phylogenomic tree by setting a midpoint root with FigTree v.1.4.3 (Rambaut, 2009).

Analysis of metagenomic read recruitment

We used anvi’o (Eren et al., 2015) v3 (available from http://merenlab.org/software/anvio/) to profile the read recruitment results following the workflow outlined by Eren et al. (2015). Briefly, we first used the program ‘anvi-gen-contigs-database’ to profile Prochlorococcus genomes, during which Prodigal v2.6.3 (Hyatt et al., 2010) with default settings identified open reading frames. We used InterProScan v5.17-56 (Zdobnov & Apweiler, 2001) and eggNOG-mapper v0.12.6 (Huerta-Cepas et al., 2016) outputs for our genes with the program ‘anvi-import-functions’ to import annotations from other databases, including PFAM (Bateman et al., 2004), and eggNOG (Jensen et al., 2008). We then used the program ‘anvi-run-ncbi-cogs’ to annotate genes with functions by searching them against the December 2014 release of the Clusters of Orthologous Groups (COGs) database (Tatusov et al., 2000) using blastp v2.3.0+ (Altschul et al., 1990). We finally used the program ‘anvi-profile’ to process the BAM file and generate an anvi’o profile database, which stored the coverage and detection statistics of each Prochlorococcus genome in the TARA Oceans data. We used ‘anvi-import-collection’ to link contigs to genomes from which they originate. Finally, the program ‘anvi-summarize’ generated a static HTML output that gave access to the mean coverage values of each genome (and individual genes within them) across metagenomes.

Operational definition of ‘population’

In the context of our study we define ‘population’ as an agglomerate of naturally occurring microbial cells, genomes of which are similar enough to align to the same genomic reference with high sequence identity as defined by the read recruitment stringency. Therefore, we assume that the isolate genomes in our study provide access to environmental populations to which they belong through the recruitment of short metagenomic reads.

Criterion for ‘detection’

Assessing the occurrence of low abundance genomes in complex data accurately can be problematic due to non-specific recruitment of short reads to regions that are conserved across multiple populations. For instance, although Prochlorococcus populations are virtually absent from the Southern Ocean (Flombaum et al., 2013), our genomes recruited up to 0.01% of the metagenomic reads from the Southern Ocean metagenomes matching to non-specific targets. To avoid high false-detection rates, we assumed that a genome was ‘detected’ in a given metagenome only if more than 50% of its nucleotide positions had at least 1X coverage.

Classification of isolate genes as ‘environmental core’ and ‘environmental accessory’

Assuming the environmental niche of a population is defined by the metagenomes in which it is ‘detected’, here we define ‘environmental core genes’ of a population as the genes that are systematically detected in its niche. In contrast, the genes that are not systematically detected within the niche of a given population represent its environmental accessory genes. Genes in a population that are classified as ‘environmental core’ given metagenomic data can be classified as ‘accessory’ given a pangenome, and vice versa. To avoid any confusion between these operationally distinct class designations, we refer to the genes classified given the metagenomic data as the ‘environmental core genes’ (ECGs), and the ‘environmental accessory genes’ (EAGs). To identify ECGs and EAGs for each genome independently, we used the anvi’o script ‘anvi-script-gen-distribution-of-genes-in-a-bin’ with the parameter ‘--fraction-of-median-coverage 0.25′. This script recovers the sum of coverage values for each gene in a given genome across all metagenomes in which the population is ‘detected’, and marks the genes that have less than 25% of the median coverage of all genes found in the genome as EAGs. We then visualized resulting gene classes using the program ‘anvi-interactive’.

Computing the pangenome, and the definition of gene clusters

The anvi’o pangenomic workflow developed for this study consists of three major steps: (1) generating an anvi’o genome database (‘anvi-gen-genomes-storage’) to store DNA and amino acid sequences, as well as functional annotations of each gene in genomes under consideration, (2) computing the pangenome (‘anvi-pan-genome’) from a genome database by identifying ‘gene clusters’, and (3) displaying the pangenome (‘anvi-display-pan’) to visualize the distribution of gene clusters across genomes, interactively bin gene clusters into logical groups, and inspect the alignment of genes in a given cluster interactively. In our study, a ‘gene cluster’ represents sequences of one or more predicted open reading frames grouped together based on their homology at the translated DNA sequence level. Gene clusters with more than one sequence may contain orthologous or paralogous sequences, or both, from one or more genomes analyzed in the pangenome. To compute the Prochlorococcus pangenome, we first generated an ‘anvi’o genomes storage database’ from the FASTA files of 31 Prochlorococcus isolate genomes using the ‘--internal-genomes’ flag. We then used the program ‘anvi-pan-genome’ with the genomes storage database, the flag ‘--use-ncbi-blast’, and parameters ‘--minbit 0.5′, and ‘--mcl-inflation 10′. This program (1) calculates similarities of each amino acid sequence in every genome against every other amino acid sequence using blastp (Altschul et al., 1990), (2) removes weak hits using the ‘minbit heuristic’, which was originally described in ITEP (Benedict et al., 2014), to filter weak hits based on the aligned fraction between the two reads, (3) uses the MCL algorithm (Van Dongen & Abreu-Goodger, 2012) to identify gene clusters in the remaining blastp search results, (4) computes the occurrence of gene clusters across genomes and the total number of genes they contain, (5) performs hierarchical clustering analyses for gene clusters (based on their distribution across genomes) and for genomes (based on gene clusters they share) using Euclidean distance and Ward clustering by default, and finally (6) generates an anvi’o pan database that stores all results for downstream analyses and can be visualized by the program ‘anvi-display-pan’.

Computing the metapangenome

Here we define ‘metapangenome’ as the outcome of the analysis of pangenomes in conjunction with the environment where the abundance and prevalence of gene clusters and genomes are recovered through shotgun metagenomes. To connect the environmental distribution patterns of genomes to the Prochlorococcus pangenome, we used the program ‘anvi-gen-samples-database’ with the genome coverage estimates reported in the summary of the anvi’o profile database for metagenomic data. To quantify the ratio of ‘environmental core genes’ (ECGs) and the ‘environmental accessory genes’ (EAGs) in each gene cluster in the resulting pangenome, we used the anvi’o program ‘anvi-script-gen-environmental-core-summary’ with default parameters. The program ‘anvi-display-pan’ visualized the Prochlorococcus metapangenome, and ‘anvi-summarize’ generated a summary of gene clusters.

Analysis of Prochlorococcus single-amplified genomes

We performed a pangenomic analysis combining the 74 SAGs and 31 isolate genomes of Prochlorococcus following the same workflow as for the isolate genomes alone. From the 74 SAGs, we then selected five phylogenetically distant ones and performed a metapangenomic analysis following the same workflow as for the isolate genomes (including the same metagenomic dataset). Our selection of few distant SAGs was intended to minimize the dilution effect due to competing read recruitment onto identical regions from multiple genomes.

Visualizations

We used the ggplot2 (Ginestet, 2011) library for R to visualize the relative distribution of genomic groups on the world map. Anvi’o performed all other visualizations, and we finalized our figures for publication using Inkscape, an open-source vector graphics editor (available from http://inkscape.org/).

Results

Environmental distribution of Prochlorococcus isolate genomes

To estimate the abundance and relative distribution patterns of the 31 Prochlorococcus isolate genomes in environmental samples, we mapped to them 30.9 billion quality-filtered metagenomic short reads from 93 TARA Oceans samples (0.2–3 µm planktonic size fraction) that cover the Atlantic Ocean, Pacific Ocean, Indian Ocean, Southern Ocean, Mediterranean Sea and Red Sea (Table S1). Prochlorococcus genomes recruited 1.68 billion reads (5.44% of the dataset) from the surface (0–15 m depth; n = 61), and the subsurface chlorophyll maximum layer (17–95 m depth; n = 32) metagenomes. The relative distribution of all Prochlorococcus genomes ranged from below the detection limit in the Southern Ocean to 24.1% in a surface metagenome from the Indian Ocean (Table S2).

In agreement with the literature, genomes from the Clade LL-II and Clade LL-III were not detected in the metagenomic dataset: although the isolation source for most LL-II/III genomes were 120 m (Rocap et al., 2002), the subsurface samples in TARA Oceans metagenomes averaged 53.7 m and never exceeded 100 m. The remaining clades displayed contrasting distribution patterns. The HL-I and HL-II genomes were enriched in surface samples, but they were geographically antagonistic: HL-I dominated in the Mediterranean Sea, while HL-II, the most abundant Prochlorococcus clade in the dataset, occurred mostly in the Indian Ocean and Red Sea (Fig. S1). Read recruitment results were also in line with previous observations suggesting temperature as one of the main drivers of distribution patterns of HL-I and HL-II (Johnson et al., 2006; Biller et al., 2014b and references therein), as 93% and 95% of the reads recruited by the HL-I and HL-II genomes originated from samples that were below and above 22°C, respectively. The LL-I and LL-IV genomes (more characteristic to the subsurface layer) were also detected in different geographic locations, but in lower proportions (Table S2). Overall, the trends observed here are largely consistent with results from previous environmental surveys and culture experiments (Johnson et al., 2006; Larkin et al., 2016), and emphasize the limited niche overlap of Prochlorococcus clades in the euphotic layer of marine systems on a large scale.

The pangenome of Prochlorococcus isolate genomes

Our pangenomic analysis of the 31 Prochlorococcus isolate genomes with a total of 60,054 genes resulted in 7,385 gene clusters. We grouped these gene clusters into five bins based on their occurrence across genomes: (1) HL + LL core gene clusters (n = 766), (2) HL core gene clusters (n = 492), (3) LL core gene clusters (n = 144), (4) singletons (i.e., gene clusters associated with a single genome; n = 2,215), and (5) other gene clusters that do not fit any of these classes (n = 3,768) (Fig. S2). The singletons and HL + LL core gene clusters corresponded to 30% and 10.4% of all clusters, respectively. This relatively small core genome is consistent with previous pangenomic investigations and supports the concept of a Prochlorococcus ‘open pangenome’ (Kettler et al., 2007). 49.1% of all clusters contained genes that were annotated with COG functions (Table S3). The functional annotation rate reached 90.5% for the HL + LL core gene clusters. In contrast, it was only 37.2% for the singletons. As the shared gene content between genomes are effective predictors of their phylogenetic relationships (Snel, Bork & Huynen, 1999; Dutilh et al., 2004), we used the distribution of gene clusters to determine the relationships among our genomes. The genomic groups that emerged from this analysis matched the six Prochlorococcus phylogenetic clades (Fig. 1). However, a noticeable difference emerged from the organization of clades based on gene clusters. Previous phylogenetic analyses using the internal transcribed spacer region (Biller et al., 2014b) placed LL genomes into polyphyletic clades (LL-I being an outlier), which was echoed by the phylogenomic analysis we performed in this study using 37 core genes (Fig. 1). In contrast, gene clusters grouped genomes primarily based on their adaptation to light regimes (Fig. 1). This result suggests that employing the whole genomic content, instead of only marker genes, may be more advantageous when the goal is to infer ecological rather than evolutionary relationships between a set of closely related genomes.

Figure 1 Organization of Prochlorococcus genomes based on shared gene clusters compared to phylogenomics.

The dendrograms on the top shows the clustering of 31 isolate genomes based on the distribution of 7,385 gene clusters recovered from the pangenomic analysis (Euclidian distance and ward clustering). The tree at the bottom organizes the same genomes based on phylogenomics using 37 concatenated core genes. Colors indicate the phylogenetic affiliations of genomes based on published literature.

Environmental core and accessory genes in Prochlorococcus isolate genomes

Genomic islands are widespread in Prochlorococcus (Coleman et al., 2006; Coleman & Chisholm, 2010) and genes from a given genome may not be found uniformly in all marine ecosystems. Besides the detection estimates at the genome level, recruiting reads from metagenomic data also provides an opportunity to investigate the occurrence and relative distribution of individual genes. We used read recruitment statistics to differentiate genes that co-occurred with the population across metagenomes from those that consistently failed to recruit reads from the environment despite the occurrence of the population. While the first group of genes is common to most cells in a given population (i.e., connected to the environment), the second group of genes occurs only in a fraction of the members of the population, or shows sporadic distribution patterns across environments (i.e., not connected to the environment). This analysis revealed 42,777 environmental core genes (ECGs) and 6,528 environmental accessory genes (EAGs) in 25 Prochlorococcus genomes (genomes from the Clade LL-II and Clade L-III were not detected in the metagenomic data, hence did not yield any estimates) (Table S3). The EAGs represented in average 13.4% (±4.65%) of all genes for each Prochlorococcus genome, exposing a non-negligible, and relatively stable portion of genes occurring only in a small subset of the cells within each population to which we had access through the genomic database and metagenomic data, consistent with previous metagenomic surveys of this lineage (Coleman & Chisholm, 2010). The synteny of most EAGs in a given genome were not random, and they mostly were clustered into hypervariable genomic islands (Fig. 2). The classification of the genes in an isolate genome based on their environmental connectivity through metagenomics offers unique insights regarding their occurrence within a population. Furthermore, this particular use of metagenomes is also essential to subsequently quantify the environmental connectivity of genes in pangenomes.

Figure 2 The gene-level detection of isolates from HL-I and HL-II in TARA Oceans metagenomes.

Visualizations describe the gene-level niche partitioning of EQPAC1 and MIT9314, two isolates from the clade HL-I and HL-II, across 93 metagenomes from TARA Oceans. For each isolate, the genes are organized based on their order in the genome, and each layer corresponds to a metagenome, which are colored based on temperature (<22 °C versus >22 °C, accordingly to Fig. 1). The most outer layer describes the environmental connectivity of each gene. Environmental core and accessory genes are colored in green and red, respectively. Genomic sections enriched in environmental accessory genes correspond to hypervariable regions.

Figure 3 The metapangenome of Prochlorococcus.

Each one of the 7,385 gene clusters contains one or more genes contributed by one or more isolate genomes. Bars in the 31 first layers indicate the occurrence of gene clusters in a given isolate genome. Gene clusters are organized based on their distribution across genomes (i.e., gene clusters that co-occur in the same group of isolates are closer to each other), and genomes are organized based on gene clusters they share using Euclidian distance and ward ordination. The three next layers describe the gene clusters in which at least one gene was functionally annotated using Pfams, EggNOGs, or COGs. Another layer describes the ratio of environmental core versus environmental accessory genes (ECGs/EAGs) within each PC. Gray areas account for the genes in genomes undetected in the metagenomic dataset.Finally, the last layer corresponds to our selections of gene clusters. The “HL + LL Core” selection corresponds to the gene clusters that contained genes from all genomes. The “LL Core” and “HL Core” selections correspond to clusters that contained genes characteristic to the LL- and HL-adapted genomes, respectively. The last selection (“Singletons”) corresponds to clusters that contained one or multiple genes from a single genome. The right-hand side section of the figure provides additional data for each isolate. The bottom rectangle displays the relative distribution of genomes across 93 metagenomes and is followed by layers that show the average distribution of each isolate in the metagenomic dataset and the phylogenetic clades to which they belong. The dendrograms on the top represents the hierarchical clustering of genomes based on the occurrence of gene clusters.

The metapangenome reveals closely related isolates with different levels of fitness

A metapangenome provides access to the environmental detection of individual genes in gene clusters, along with the ecological niche boundaries of individual genomes. The Prochlorococcus metapangenome revealed differences within the members of the Clade HL-II with respect to their rate of detection in the environment (Fig. 3; see the interactive version at the URL http://anvi-server.org/p/JNlBAB). Interestingly, the organization of genomes in HL-II based on gene clusters matched their detection gradient within their niche, with the least abundant and the most abundant genomes in the metagenomic data being at the two extremes of the cluster that described the Clade HL-II (Fig. 3, Table S2). We tentatively grouped the HL-II genomes into three sub-groups based on their abundance in the metagenomic dataset: HL-II-Low (n = 3) with an average relative abundance of 0.037%, HL-II-Medium (n = 10) with an average relative abundance of 0.14%, and HL-II-High (n = 4) with an average relative abundance of 0.5%. Based on this grouping, HL-II-High genomes were 13.5 times more abundant in the environment on average compared to HL-II-Low genomes, despite being closely related enough to be described in the same phylogenetic group for HL. In light of this observation, we investigated whether the differentially distributed gene clusters could identify the functional basis of the apparent change in fitness. Noticeably, the HL-II-Low genomes were lacking gene clusters that resolve to DNA repair (DNA ligase; 3-methyladenine DNA glycosylase; DEAD DEAH box helicase) compared to the HL-II-High genomes (Table S3). All 31 isolates carried DNA repair genes, as it is a critical protection mechanism towards light induced damages occurring in the surface layer of marine systems (Jeffrey et al., 1996); however, HL-II-High genomes carried a unique set of DNA repair genes that were missing from HL-II-Low genomes. Also missing from the HL-II-Low genomes were gene clusters corresponding to enzymes of the cupin superfamily, the fructose-bisphosphate aldolase class II, glutamine amino transferase, PAP fibrilin, a metal-binding protein, and 25 gene clusters to which we could not assign a function. The metapangenome provided access to genomic features that may explain the functional basis of such variation of fitness between closely related members of the HL-II group. Assuming that an increased relative abundance in the environment is equivalent to increased fitness, characterization of the genomic features that contribute to these differences, especially those of unknown functions, warrants further study.

Genes and functions connect the hypervariable genomic islands of Prochlorococcus populations

We then turned our attention to the key contribution of our metapangenomic workflow; the environmental connectivity of the pangenome as defined by the proportion of ECGs and EAGs found in each gene cluster. The percentage of EAGs from genomes that occurred in our metagenomic data differed markedly between the HL + LL core gene clusters (4.31%), LL core gene clusters (0.28%), HL core gene clusters (12.4%), and singletons (66%) (Fig. 3; Table S3). More than an order of magnitude difference between the ratio of ECGs to EAGs among the LL and HL core gene clusters suggests that, given the available isolate genomes, Prochlorococcus genes characteristic to low-light regime may be more stable than those characteristic to high-light regime. These results also indicate that genes present in all isolate genomes (HL + LL core) were maintained in a large fraction of the cells in populations we investigated, while those that are specific to a single isolate largely occurred in smaller number of cells in the environment and remained below our detection limit. Exceptions to low number of EAGs in HL + LL core were gene clusters #33, #44 and #431 (see Table S3). The percentage of EAGs for these gene clusters in HL isolates were 100%, 95.2% and 95.2%, and their functions resolved to ‘nucleotide sugar epimerase’, ‘udp-glucose 6-dehydrogenase’ and ‘mannose-1-phosphate guanylyltransferase’, respectively. In contrast, these gene clusters contained only ECGs in the LL isolates (Table S3). Sugar uptake by Prochlorococcus has been observed in both culture and in situ (Gomez-Baena et al., 2008; Muñoz Marín et al., 2013; Muñoz-Marín et al., 2017) and this process can support the growth of Prochlorococcus populations in the surface ocean (Moisander et al., 2012). The occurrence of multiple sugar metabolism genes in every HL isolate that are absent in almost all metagenomes poses an interesting conundrum.

To investigate whether this could be due to a cultivation bias that selects for members from these populations with a certain set of sugar utilization genes, we analyzed 74 single amplified genomes (SAGs) from a study by Kashtan et al. (2014) (Table S4). Our analysis revealed that these gene clusters also occurred in a large number of SAGs (75.7% to 81.1%) (Table S4). Most interestingly, metapangenomic analysis of SAGs using the same metagenomic dataset and bioinformatics workflow we used for the isolates also revealed that all genes in these gene clusters were EAGs (Table S4), consistent with our observations in the HL isolates, and ruling out the ‘cultivation bias’ hypothesis. Yet these results left us with a puzzling observation as we have identified Prochlorococcus gene clusters widespread in both isolate genomes and SAGs of the HL clades with genes rarely detected in the surface oceans and seas. Methodological differences could explain the conflict between the high prevalence of these gene clusters across genomes in the pangenome and the low detection of each gene in them across metagenomes: gene clusters are formed based on homology between amino acid sequences (Tettelin et al., 2005), hence can contain genes with relatively low sequence similarity, while metagenomic read recruitment is done at the DNA sequence-level, and is more stringent.

Notably, genes in clusters #33, #44 and #431 occurred in hypervariable genomic islands of the isolates and SAGs (Fig. 4, Tables S3 and S4), and as a result are surrounded by other EAGs that are not part of the Prochlorococcus core genome. To the best of our knowledge this is the first time the Prochlorococcus core pangenome is linked to hypervariable genomic islands, indicating that core functionalities of this major lineage associated with sugar metabolism are maintained in a variety of versions within each population. Finally, analyzing the functionality of all EAGs led us to expose a prevalent role of sugar metabolism in hypervariable genomic islands beyond the three core gene clusters (Fig. 4 and Fig. S3). Briefly, functions such as udp-glucose 4-epimerase, dTDP-4-dehydrorhamnose 3,5-epimerase, dTDP-4-dehydrorhamnose reductase, dTDP-glucose 4-6-dehydratase, GDP-mannose 4,6-dehydratase and glucose-1-phosphate cytidylyltransferase were dominated by EAGs and occurred mostly in hypervariable genomic islands of the HL populations (Table S5). Overall, our analyses suggested a high rate of gene diversification traits for sugar metabolism in Prochlorococcus that may be contributing to the remarkable fitness of this group in the surface ocean.

Figure 4 Prevalence of sugar utilization in Prochlorococcus hypervariable genomic islands.

(A) describes the 25 most environmental accessory functions identified in Prochlorococcus isolates defined by unusually high ratio of EAGs. (B) and (C) display the coordinates of genes corresponding to the 25 most environmental accessory functions across five isolates genomes and five SAGs of Prochlorococcus, respectively (red in the outer layers). Inner layers correspond to the 93 TARA Oceans metagenomes, organized by geographic regions similarly to Fig. 2. For each metagenome, black sections correspond to well covered genes while white sections correspond to genes with no read recruitment.

Discussion

The quantity of data in genomic databases and metagenomic surveys is increasing rapidly thanks to the advances in biotechnology and computation. Metapangenomes take advantage of both genomes and metagenomes to link two important endeavors in microbiology: inferring the relationships between isolate genomes through identifying the core and accessory genes they harbor de novo, and investigating the relative distribution of microbial populations and individual genes in the environment through metagenomics.

Our metapangenomic workflow has similarities to the method described in a recently introduced metagenomics pipeline by Nayfach et al. (2016), as both efforts offer solutions to expand conventional analyses of pangenomes by not only estimating the abundance and distribution of gene clusters in the environment, but also linking them to the distribution patterns of microbial populations. In addition to this shared goal, our approach provides a flexible starting point with project-specific genomic databases (rather than pre-computed references), and includes a comprehensive visualization strategy to summarize metapangenomes.

The Prochlorococcus metapangenome revealed subtle distribution gradients among isolates that belonged to the same phylogenetic clade, and exposed differentially occurring gene clusters that could be related to genomic traits affecting the fitness among closely related members. It also revealed gene clusters that occurred in every isolate genome and in most single-cell genomes but were largely missing in the environment, exposing a core genome connecting hypervariable genomic islands of distinct Prochlorococcus phylogenetic clades. Interestingly, these gene clusters were biased towards sugar utilization. Variable genomic islands of Prochlorococcus among co-occurring cells (Coleman et al., 2006) have previously been linked to the resistance of viral infections (Avrani et al., 2011). Our findings here suggest that high sequence diversification among genes involved in sugar metabolism may be beneficial for Prochlorococcus populations, which should be further addressed. In addition, gene clusters revealed that at least some of the genes in Prochlorococcus genomic islands represent common functions with high rate of intra-population diversity at the DNA-level, rather than recent horizontal transfers from other lineages. These observations contribute to the ongoing debate on the origin, evolution and ecological role of hypervariable genomic islands within microbial populations (Hacker & Carniel, 2001; Coleman et al., 2006; Wilhelm et al., 2007; Juhas et al., 2009; Fernández-Gómez et al., 2012; Vineis et al., 2016). In addition to these novel insights, the parallels in our findings and the extensive literature on Prochlorococcus emphasizes the potential of metapangenomics to facilitate the recovery of key insights from novel and less studied microbial populations, including those with no cultured representatives.

The vast majority of isolate and single-amplified genomes contain only a subset of the complete set of genes microbial populations maintain within their niche boundaries (Parkhill et al., 2000; Coleman et al., 2006; Juhas et al., 2009; Coleman & Chisholm, 2010). Metagenomic data make it possible to classify genes in genomes based on their occurrence in the environment. However, metagenomic short read recruitment alone does not provide access to genes that are lacking in available genomes, even if they may be critical for the functioning of the populations they originate. Characterizing all accessory genes of a given population in the environment is challenging due to the limited coverage of the environmental metagenomes and genomic databases. These limitations require careful interpretations of the observations that emerge from the metapangenomic workflow and awareness that complete understanding of the accessory genes of the environment may require additional efforts (Kashtan et al., 2014).

Conclusion

Here we developed novel software solutions and analytical tools within the open-source software platform anvi’o to create and study metapangenomes with interactive visualization and inspection capabilities. Our analysis of the Prochlorococcus metapangenome revealed a small number of gene clusters that may be linked to subtle fitness trends among very closely related members of this group, and displayed inter-connectivity of hypervariable genomic islands across multiple clades. Our findings suggest that metapangenomes can provide highly resolved linkage between core and accessory genes of microbial populations and the environment, for any taxon and biome for which genomic and metagenomic data are available, and can provide experimental targets to explore the functional basis of niche partitioning and fitness. Besides isolate and single-cell genomes, this strategy can also employ metagenome-assembled genomes, and be used to study questions in the context of biotechnology or medicine.

Supplemental Information

Figure S1 The distribution of isolates from HL-I and HL-II in TARA Oceans metagenomes

World maps describe the cumulative relative distribution of Prochlorococcus isolates from the clades HL-I (3 genomes) and HL-II (17 genomes) across 61 surface metagenomes. The size and color of dots varies as a function of relative distributions and temperature range (<22 °C versus >22 °C), respectively.

Click here for additional data file.

Figure S2 The pangenome of Prochlorococcus

Each one of the 7,385 gene clusters contains one or more genes contributed by one or more isolate genomes. Bars in the 31 horizontal layers indicate the occurrence of gene clusters in a given isolate genome. Gene clusters are organized based on their distribution across genomes (i.e., gene clusters that co-occur in the same group of isolates are closer to each other), and genomes are organized based on gene clusters they share using Euclidian distance and ward ordination. The “HL + LL Core” selection corresponds to the clusters that contained genes from all genomes. The “LL Core” and “HL Core” selections correspond to gene clusters that contained genes characteristic to the LL- and HL-adapted genomes, respectively. The last selection (“Singletons”) corresponds to clusters that contained one or multiple genes from a single genome.

Click here for additional data file.

Figure S3 Prevalence of sugar utilization in Prochlorococcus hypervariable genomic islands

The figure displays the coordinates of genes corresponding to the 25 most environmental accessory functions across isolates genomes Prochlorococcus (red in the outer layers). Inner layers correspond to the 93 TARA Oceans metagenomes, organized by geographic regions similarly to Fig. 2. For each metagenome, black sections correspond to well covered genes while white sections correspond to genes with no read recruitment. Genomes are organized based on gene clusters similarly to Fig. 3.

Click here for additional data file.

Table S1 Summary of genomes and metagenomes

Summary of 31 Prochlorococcus genomes and 93 metagenomes from the TARA Oceans project.

Click here for additional data file.

Table S2 Genomic detection in the environment

Reads recruitments, detection and relative distribution of 31 Prochlorococcus genomes in 93 metagenomes from the TARA Oceans project.

Click here for additional data file.

Table S3 Metapangenomics summary

Summary of the metapangenomic analysis of Prochlorococcus isolates. The table describes the functionality and environment connectivity of genes identified in the 31 Prochlorococcus isolate genomes, and links each gene to a gene cluster in the Prochlorococcus pangenome.

Click here for additional data file.

Table S4 Metapangenomics summary when including SAGs

Summary of the pangenomic and metapangenomic analyses of Prochlorococcus isolates and SAGs. The table describes the pangenomic analysis of 31 Prochlorococcus isolate genomes and 74 Prochlorococcus single cell genomes (SAGs). The table also describes the metapangenome of five SAGs, which includes the functionality and environment connectivity of genes and links each gene to a gene cluster in the corresponding Prochlorococcus pangenome.

Click here for additional data file.

Table S5 Environmental connectivity of functions

Summary of the environmental connectivity of functions identified in the 31 Prochlorococcus isolates. The table also links the environmental connectivity of functions to the different clades of Prochlorococcus.

Click here for additional data file.

We thank Bana Jabri, Sean Crosson, Ryan J. Newton, Maureen L. Coleman, Bas Dutilh, Loïs Maignien, Julie Reveillaud, Michael D. Lee, and the members of the Meren Lab for helpful discussions. We are also grateful to our anonymous reviewers for scrutinizing our work, Özcan C. Esen for his technical insights and help, and Hilary G. Morrison for her guidance to improve our manuscript. Finally, we are indebted to the scientists who made this study possible by generating the genomes and metagenomes, and making them publicly available.

Additional Information and Declarations

Competing Interests

Author Contributions

Data Availability

A. Murat Eren is an Academic Editor for PeerJ.

Tom O. Delmont and A. Murat Eren conceived and designed the experiments, performed the experiments, analyzed the data, contributed reagents/materials/analysis tools, wrote the paper, prepared figures and/or tables, reviewed drafts of the paper.

The following information was supplied regarding data availability:

The TARA Oceans metagenomes are publicly available through the European Bioinformatics Institute (accession IDs ERP001736) at https://www.ebi.ac.uk/metagenomics/projects/ERP001736.

We also made available:

(1) Prochlorococcus isolate genomes and SAGs https://doi.org/10.6084/m9.figshare.5447221.v1;

(2) the anvi’o database files and the static HTML summary output for Prochlorococcus isolate genomes across TARA Oceans metagenomes

https://doi.org/10.6084/m9.figshare.5447224;

(3) the metapangenome of Prochlorococcus isolates https://doi.org/10.6084/m9.figshare.5447227; an extended pangenome of Prochlorococcus isolates and SAGs https://doi.org/10.6084/m9.figshare.5447230;

(4) and the metapangenome of Prochlorococcus SAGs https://doi.org/10.6084/m9.figshare.5447233.

The URL https://anvi-server.org/merenlab/prochlorococcus_metapangenome serves an interactive version of the metapangenome of Prochlorococcus isolates.

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
