# Peer review of "Linking pangenomes and metagenomes: the Prochlorococcus metapangenome"

_PeerJ, doi:10.7717/peerj.4320_

## Round 0.1 · original submission · Minor Revisions

Pleas address all comments of the two reviewers.

Reviewer 1 ·

Basic reporting

The Authors show how pangenomes and metagenomes can be linked and provide proof-of-concept of how this metapangenomics provides unique insights.

The English should be improved to ensure text is clearly understood. For example:
1/ Line 27 to 32. In the abstract, the authors give two statements, “Rapidly growing number of ... …of populations across microbial genomes.” The first statements is a general statement that is followed by a second statement that is supposed to provide more clarity of which aspects of the general statement is the key focus of this manuscript. However, the current phrasing makes comprehension difficult.
2/ Line 65 to 68. Also rephrase these statements to make comprehension easy.
3/ Line 274. Rephrase these statements to make comprehension easy.
4/ Line 411 to 415. Rephrase these statements to make comprehension easy.
5/ The entire document needs to be proofread.

Most of the references used are from the Nature Journal but some references are old and newer published manuscripts with impacting findings have not been included. For example:
1/ Line 69: include after reference (“Lorenz & Eck 2005; Thies, Stephan, et al. "Metagenomic discovery of novel enzymes and biosurfactants in a slaughterhouse biofilm microbial community." Scientific reports 6 (2016): 27035.)
2/ Line 70: include after reference (“Tringe et al., 2005; Al-Amoudi, Soha, et al. "Metagenomics as a preliminary screen for antimicrobial bioprospecting." Gene 594.2 (2016): 248-258)
3/ line 71-72: include after reference (“Tyson et al., 2004; Haroon, Mohamed F., et al. "A catalogue of 136 microbial draft genomes from Red Sea metagenomes." Scientific data 3 (2016): 160050; Delmont et al., 2017)

Overall, I commend the authors for the thorough data analyses and on conciseness of style of writing. If there is a weakness it merely is with respect to making comprehension easier (as I have noted above).

Experimental design

Research question well defined and meaningful.

Validity of the findings

Conclusion are well stated, linked to original research question & limited to supporting results.

Reviewer 2 ·

Basic reporting

I am putting my entire review in this section, as nearly all, if not all, of my comments are related to basic reporting.

Overall:

This is a nice contribution by Delmont and Eren describing the utility of a new software pipeline in the existing Anvi’o tool, along with a few new insights into Prochlorococcus ecology. The pipeline links genes from isolate genome sequences to their abundances in the environment via metagenomic read mapping, and it can identify specific genes (or protein clusters) that exist in isolate genomes but may be very uncommon in the environment. I have few, if any, scientific criticisms, but I found a lot of the text confusing, mostly due to undefined terminology and some long, confusing sentences. I think that this is a relatively straightforward study that would benefit from some streamlining of the text. Specific comments are below.

Abstract:

-The abstract is somehow both well written and deeply confusing. Please simplify the language. I am left not really understanding what the main question(s), methods, and results are. I know intuitively what both metagenomics and pangenomics (or at least pangenomes) are, but it would be helpful for the authors to explain these terms in the context of how they were considered for this study. Is the sentence starting with “While pangenomics offers …” meant to define both terms? If so, please restructure it along the lines of “The pangenome of a population (or genus?) consists of both core (shared) and accessory genes and genomic features …” or however you want to define it, and please similarly define metagenomics and its use in this study. If metagenomics is being used for abundance estimates (abundance estimates of what – SNPs, populations, and/or genes within pangenomes?), then consider calling it something more direct (metagenomic abundance estimates?). To me, the term metagenomics primarily evokes community predicted functional profiling and/or population genome assembly and metabolic reconstruction. Even though I have personally also made abundance estimates from metagenomes in much the same way as the authors, that would not be what first comes to mind. [edit: that is almost exactly how this is described by the authors in ln 69-74, so this needs to be much more clear in the abstract]

-Along those lines, I do not think of metagenomics and pangenomics to be inherently different. I would assume that metagenomics could be used both to define the pangenome (i.e., to find core and accessory genes in metagenomic assemblies) and to determine the abundances of subpopulations and/or specific genes or regions of the pangenome (through read mapping to metagenomic assemblies, SAGs, isolate genomes, or any combination of the above). Which, if any, of these possibilities apply to this study is not clear.

-Metapangenomics is not defined, and I do not find it to be a helpful term. Consider removing it from the manuscript, including the title. Based on my reading of the abstract alone, it looks like metapangenomics is meant to describe an abundance-informed pangenome, and if so, why not call it something like that, with some useful meaning in the term itself?

Main Text:

-ln 32: complementary

-ln 38-42: What does this sentence mean? Consider splitting it into two sentences. How does a metapangenome correlate with something? What is it correlating with? I am not sure that “traits” is an appropriate word – are the authors describing core and accessory genes here? What are “sub-clade demarcations”? How would these results differ from phylogenetic analyses (wouldn’t phylogenetics by definition separate sub-clades?)? Do the authors mean some specific phylogenetic analyses that are typically performed at a coarser resolution? If so, please provide more context on the phylogenetic analyses.

Main text:

-ln 54: “have been” should be “has been”

-Throughout: consider changing “shared” to “core” in the context of core genes across pangenomes, as this is more common in the literature and therefore more intuitive. If the authors mean shared among some but not all populations, then that should be explicitly mentioned, but I assume that they mean core genes shared among all populations.

-ln 74-76: What does this mean? Wouldn’t metagenomic assembly + read mapping do that too? It might help if you explain the particular utility of isolate sequences here.

-ln 77-78 (or earlier): Please define functional traits in the context of a microbial genome or pangenome. I think that you also mean isolate genomes here, so please change the end to “… mapping of closely related isolate genomes.”

-ln 80-83: This sentence is a bit long and confusing and can probably be broken down. The authors can rework it for clarification as they see fit, but here are some examples of what I find confusing: What are “well-established practices in pangenomics”? Please give a few examples. What are “emerging opportunities from metagenomic data”? Is this just using metagenomic read mapping for abundance data? If so, just say that. What is a genome-centric framework (I assume that this involves the use of closed, isolate genomes), and how does that differ from what you would get from a combination of metagenomic assembly and binning to identify populations + read mapping across a number of metagenomes to get abundance estimates? What are “pangenomic traits” and how do you define which ones are “key”? Are “key” traits just those that are linked to “niche partitioning” and “population fitness”, and if so, how do you determine that?

-ln 85: please state exactly what you mean by integrating pangenomic and metagenomic data. Again, what is “pangenomic data” and what is “metagenomic data”? Are there better terms for these types of data in this context, for example, “… integrating population pangenomes from multiple isolate genome sequences with their abundance profiles across environmental samples from metagenomic read mapping?” That might not even be correct, but the point is that I do not understand.

-I think that the focus of both the abstract and introduction and maybe even the title should be on the need for and development of this software pipeline, as that seems to be the key novel result of the study, tested on Prochlorococcus as an example, right? [[later edit: wait, but the tool is Anvi’o, which is fabulous but not new; please use the introduction to very clearly walk the reader through what is known vs. what is new in this study, both in terms of the visualization software pipeline and the Prochlorococcus biology]]. It seems dangerous to imply that metagenomics has never been used to identify the ecological niches of specific subpopulations (for example, the Banfield lab has worked in that general area, at least in AMD systems; how would isolate pangenomes add further information in that context?) and much safer to say that your software and visualization pipeline can help to identify and show these differences more clearly.

-ln 93: How many genomes? That number seems important if all of these genomes are going into your downstream analyses.

-ln 95: Were these 16S rRNA gene amplicon surveys or otherwise not metagenomic studies? That seems like a worthwhile point of clarification to help make the case for the current study that links isolate genomes to metagenomic data.

-ln 96: dynamics plural

-ln 97-98: Correlations between the “genomic traits” of isolates … are these just groups of genes that correlate with environmental variables? Were there correlations to variables other than HL and LL? If so, maybe call these “other environmental variables.”

-ln 98-104: This is a long sentence, and I got lost halfway through. Do “these two groups” refer to the core and accessory genes? How many metagenomes? What does “independently” mean here, and what is “their differential occurrence”? Does “in Prochlorococcus populations” mean in the same 12 as at the beginning of the sentence? If so, change to “in the 12 Prochlorocuccus populations,” otherwise define these populations. I don’t understand the last part of the sentence at all. Maybe summarize these three studies in three separate sentences and explain which part(s) of each are being included in the current analyses, and then explain clearly how the current study will expand on what is already known from these previous studies.

-ln 104-105: This seems like an important distinction. To this point, the general implication is that this is the first time that anybody has thought of exploring niche partitioning in pangenomes or metagenomes, yet here the authors say that the difference is that previous studies have not had resolution at the level of protein clusters. Again, please dispense with the fancy sentences and terms and use the introduction to tell the reader what has been done in the past that is relevant, both in terms of biology and visualization/software, and then explicitly state what knowledge gap will be filled by this study. For example, it would be useful to explicitly say why monitoring protein clusters is useful.

-ln 106: Again, what are “pangenomic traits”? Maybe I am just stuck on “traits” as an ecological term and the authors just mean similarities and differences across populations?

-ln 106: How do these 31 Prochlorococcus isolates relate to the 12 (or more?) populations described from previous studies above?

-ln 108: Please give an exact number for billions

-ln 110: Define ecological niche; is this just HL vs. LL here?

-ln 109-115: These are results that do not belong in the Introduction. Consider either reworking to frame these as hypotheses (or similar) that will be explored in this study, or remove this.

-ln 113 and 117: These are the first mentions of SAGs. This seems to be of abstract-level importance in how you are defining your pangenomes (i.e., a combination of SAGs and isolates). Or am I not understanding how your pangenomes were defined? After reading more of the results, I do not see much in the way of SAGs there, so how did you decide when to use isolates and when to use SAGs in your analyses? SAGs are presumably less complete genomes, so if a particular gene is not detected in a SAG, it does not necessarily mean that it is actually absent. The authors know this, I am sure, but if this is part of the rationale for using only isolates for some of the analyses, it should be mentioned.

-I think that Anvi’o can also be called out specifically in the Abstract and/or Introduction. It is not clear to me what aspects of the Anvi’o workflow are new in this study, though the Introduction suggests that this is a novel pipeline. Based on the text to this point, I was expecting the presentation of a novel workflow, and this needs to be made more clear. I think that a paragraph in the Intro with Anvi’o background would be appropriate – how has Anvi’o been used in the past, and what specifically is the new application here? It seems like more than just plugging new data into the software, so maybe a flowchart figure would help? There is a section of the methods dedicated to this, which is good, but I wonder if at least some of that should be moved to the main text, given that the pipeline is one of the key outputs of the study and not just an ancillary method.

-ln 134: Has phylogenomics not been done on these 31 genomes before?

-ln 176-199: Is this the new part? If so, you could start with something along the lines of “The Anvi’o pangenomic workflow developed for this study consists of …”

-ln 179: What is a “genome of interest”? Is this just every genome that will be considered for a given analysis, i.e., 31 Prochlorococcus isolates for this study? [I see later that this is the case, so please rephrase to make this more clear]

-ln 234 and 264: What about the SAGs?

-ln 234-243: Has this been done already for Prochlorococcus in any of the TARA Oceans publications? I would guess so, but maybe not all 31 isolates were included. It would be worth clarifying what parts of this analysis are new vs. what just needed to be done again here to feed into the Anvi’o pipeline.

-ln 244-256: These specific clades have not been described anywhere. I realize that a description of each could get tedious, but is there something general that you could say along the lines of, e.g., “All LL lineages come from low-light niches and include subclades I-IV defined by x, y, z” Otherwise, the description of these clades is not particularly useful. The figures just say that these are “literature-defined” lineages, which is fine, but the authors could briefly elaborate on these clade distinctions in the text.

-ln 277-285: Cool!

-ln 298-299: ECGs and EDGs -- do we really need more acronyms? I saw these again a couple of pages later and had to dig back to this section to remind myself of what they are. [and again when I came back to the manuscript after a break] When these acronyms appear again a couple pages later (ln 356), the next sentence has four different acronyms occurring eight times …

-ln 313: Okay, metapangenomics is finally defined! I still don’t fully understand the utility of this term. Maybe it is just me, but I do not find the introduction of new terms and acronyms in nearly every new manuscript in this field to be helpful.

-ln 313-320: What is the result here? The result cannot just be the figure; there has to be some interpretation or guidance for what the reader should be seeing.

-ln 354-356: This seems like an important contribution, and it is buried near the end of the Results.

-ln 366-367: Change to plural

-ln 370-371: Please rephrase this sentence for clarity. What is “they”?

-ln 420: Is this really only a little effort?!

-ln 464-466: Okay, the authors have confirmed the obvious application that I mentioned above, which is that this can also be applied to metagenome-assembled genomes. Why were those not considered here? I don’t think that this is a hole-in-the-paper offense, but I am puzzled, as it seems like a relatively easy addition that would boost the size of the available pangenome significantly.

Figures:

These are nice. If the authors insist on keeping the ECG and EDG acronyms, please define them in each figure legend.

Experimental design

See above (Basic reporting section) for a few specific comments related to better defining the research question and knowledge gap(s) filled by this study in the Introduction.

Validity of the findings

No comment

Additional comments

No comment

---

## Round 0.2 · accepted · Accept

Both reviewers confirm that the contribution merits publication in PeerJ and I concur.

Reviewer 1 ·

Basic reporting

The manuscript has been extensively reviewed and English use is clear and unambiguous.

Experimental design

In the current form, the manuscript results is relevant to the hypothesis and it is now clear how the work fits into the broader field of knowledge.

Validity of the findings

The findings are appropriately stated, and connected to the original question investigated.

Reviewer 2 ·

Basic reporting

no comment

Experimental design

no comment

Validity of the findings

no comment

Additional comments

I thank the authors for addressing my concerns and making the manuscript much more clear. I have no further comments.

---

## Author Rebuttal · Round 0.2

A. Murat Eren, Ph.D.
ASSISTANT PROFESSOR, DEPARTMENT OF MEDICINE

**Knapp Center for Biomedical Discovery**
900 E. 57th Street, Mailbox 9, Room 9118, Chicago, IL 60637
P: +1-773-702-5935 / F: +1-773-702-2281 / meren@uchicago.edu

11 December, 2017

Dear Vladimir Bajic,

We would like to thank you and our reviewers very much for evaluating our study.

Among other important points, the comments from our reviewers helped us realize that we needed to improve the readability of our study by simplifying our terminology and clarifying our definitions.

We believe our revised version offers a better reading experience. One of the major changes that were not specifically requested by our reviewers, yet we felt that it was necessary as a part of simplifying the language in our study was the renaming of 'protein clusters' to 'gene clusters'. For this, we consulted with the community, and made sure the use of 'gene clusters' was acceptable by other scientists who are working with pangenomes:

https://github.com/merenlab/anvio/issues/644

We also included a URL in our methods section that leads to a detailed workflow that extends the descriptions and parameters of programs we used in our study.

The following is our point-by-point responses (in blue) to reviewer comments (in black). Our submission also includes the manuscript file with tracked changes.

We hope that you and our reviewers will find our revised manuscript satisfactory.

A. Murat Eren

[Figure]

A. Murat Eren, Ph.D.

ASSISTANT PROFESSOR, DEPARTMENT OF MEDICINE

**Knapp Center for Biomedical Discovery**
900 E. 57th Street, Mailbox 9, Room 9118, Chicago, IL  60637
P: +1-773-702-5935 / F: +1-773-702-2281 / meren@uchicago.edu

# Reviewer #1 (Anonymous)

Basic reporting

The Authors show how pangenomes and metagenomes can be linked and provide proof-of-concept of how this metapangenomics provides unique insights.

The English should be improved to ensure text is clearly understood. For example:

1/ Line 27 to 32. In the abstract, the authors give two statements, "Rapidly growing number of ... …of populations across microbial genomes." The first statements is a general statement that is followed by a second statement that is supposed to provide more clarity of which aspects of the general statement is the key focus of this manuscript. However, the current phrasing makes comprehension difficult.

We thank the reviewer for their input regarding the language issues. We modified our text carefully to improve its clarity. The particular sentence in the abstract the reviewer pointed out now reads as follows:

> "*Pangenomes offer detailed characterizations of core and accessory genes found in a set of closely related microbial genomes, generally by clustering genes based on sequence homology. In comparison, metagenomes facilitate highly resolved investigations of the relative distribution of microbial genomes and individual genes across environments through read recruitment analyses. Combining these complementary approaches can yield unique insights into the functional basis of microbial niche partitioning and fitness, however, advanced software solutions are lacking.*"

2/ Line 65 to 68. Also rephrase these statements to make comprehension easy.

Done. It now reads:

> "*Shotgun metagenomics, the sequencing of DNA directly extracted from the environment (Handelsman et al., 1998), allows the study of microbial communities without the need for cultivation.*"

3/ Line 274. Rephrase these statements to make comprehension easy.

We included relevant citations to improve the clarity of the statements. It now reads:

> "*As the shared gene content between genomes are effective predictors of their phylogenetic relationships (Snel, Bork & Huynen, 1999; Dutilh et al., 2004), we used the distribution of gene clusters to determine the relationships among our genomes. The genomic groups that emerged from this analysis matched the six Prochlorococcus phylogenetic clades (Figure 1).*"

4/ Line 411 to 415. Rephrase these statements to make comprehension easy.

Done. It now reads:

[Figure]

A. Murat Eren, Ph.D.

ASSISTANT PROFESSOR, DEPARTMENT OF MEDICINE

**Knapp Center for Biomedical Discovery**
900 E. 57th Street, Mailbox 9, Room 9118, Chicago, IL 60637
P: +1-773-702-5935 / F: +1-773-702-2281 / meren@uchicago.edu

*"The quantity of data in genomic databases and metagenomic surveys is increasing rapidly thanks to the advances in biotechnology and computation. Metapangenomes take advantage of both genomes and metagenomes to link two important endeavors in microbiology: inferring the relationships between isolate genomes through identifying the core and accessory genes they harbor de novo, and investigating the relative distribution of microbial populations and individual genes in the environment through metagenomics."*

5/ The entire document needs to be proofread.

We thank the reviewer for their patience. The document now has been read and corrected by Hilary G. Morrison, a senior scientist at the Marine Biological Laboratory, whose native language is English.

Most of the references used are from the Nature Journal but some references are old and newer published manuscripts with impacting findings have not been included. For example:

1/ Line 69: include after reference ("Lorenz & Eck 2005; Thies, Stephan, et al. "Metagenomic discovery of novel enzymes and biosurfactants in a slaughterhouse biofilm microbial community." Scientific reports 6 (2016): 27035.)

We thank the reviewer for helping us improve the coverage of the literature independent of the journal impact factor.

2/ Line 70: include after reference ("Tringe et al., 2005; Al-Amoudi, Soha, et al. "Metagenomics as a preliminary screen for antimicrobial bioprospecting." Gene 594.2 (2016): 248-258)

Done.

3/ line 71-72: include after reference ("Tyson et al., 2004; Haroon, Mohamed F., et al. "A catalogue of 136 microbial draft genomes from Red Sea metagenomes." Scientific data 3 (2016): 160050; Delmont et al., 2017)

Done.

Overall, I commend the authors for the thorough data analyses and on conciseness of style of writing. If there is a weakness it merely is with respect to making comprehension easier (as I have noted above).

We thank the reviewer for their interest in our study, and their time for helping us improve the readability of the manuscript.

Experimental design

Research question well defined and meaningful.

Validity of the findings

Conclusion are well stated, linked to original research question & limited to supporting results.

[Figure]

A. Murat Eren, Ph.D.

ASSISTANT PROFESSOR, DEPARTMENT OF MEDICINE

**Knapp Center for Biomedical Discovery**

900 E. 57th Street, Mailbox 9, Room 9118, Chicago, IL  60637

P: +1-773-702-5935 / F: +1-773-702-2281 / meren@uchicago.edu
* * *
# Reviewer #2 (Anonymous)

Basic reporting

I am putting my entire review in this section, as nearly all, if not all, of my comments are related to basic reporting.

Overall:

This is a nice contribution by Delmont and Eren describing the utility of a new software pipeline in the existing Anvi'o tool, along with a few new insights into Prochlorococcus ecology. The pipeline links genes from isolate genome sequences to their abundances in the environment via metagenomic read mapping, and it can identify specific genes (or protein clusters) that exist in isolate genomes but may be very uncommon in the environment. I have few, if any, scientific criticisms, but I found a lot of the text confusing, mostly due to undefined terminology and some long, confusing sentences. I think that this is a relatively straightforward study that would benefit from some streamlining of the text. Specific comments are below.

*We thank the reviewer for their interest in our study, and for their suggestions.*

*Their comments encouraged us to put more attention into our text, and make it more accessible to its readers. We simplified our terminology, added clearer definitions throughout, and split complex sentences into simpler ones. We also consulted with our colleagues to improve the overall language of the manuscript.*

Abstract:

-The abstract is somehow both well written and deeply confusing. Please simplify the language. I am left not really understanding what the main question(s), methods, and results are. I know intuitively what both metagenomics and pangenomics (or at least pangenomes) are, but it would be helpful for the authors to explain these terms in the context of how they were considered for this study. Is the sentence starting with "While pangenomics offers …" meant to define both terms? If so, please restructure it along the lines of "The pangenome of a population (or genus?) consists of both core (shared) and accessory genes and genomic features …" or however you want to define it, and please similarly define metagenomics and its use in this study. If metagenomics is being used for abundance estimates (abundance estimates of what – SNPs, populations, and/or genes within pangenomes?), then consider calling it something more direct (metagenomic abundance estimates?). To me, the term metagenomics primarily evokes community predicted functional profiling and/or population genome assembly and metabolic reconstruction. Even though I have personally also made abundance estimates from metagenomes in much the same way as the authors, that would not be what first comes to mind. [edit: that is almost exactly how this is described by the authors in ln 69-74, so this needs to be much more clear in the abstract]

*We thank the reviewer for pointing this out, which also was a concern of the Reviewer #1. We largely rewrote the abstract to improve its clarity. Among many changes in the abstract, we included a more specific definition of pangenomes, and specified our use of metagenomes by*

[Figure]

A. Murat Eren, Ph.D.

ASSISTANT PROFESSOR, DEPARTMENT OF MEDICINE

**Knapp Center for Biomedical Discovery**
900 E. 57th Street, Mailbox 9, Room 9118, Chicago, IL 60637
P: +1-773-702-5935 / F: +1-773-702-2281 / meren@uchicago.edu

mentioning their relevance to investigate the relative distribution of genomes in the environment as the reviewer suggested.

-Along those lines, I do not think of metagenomics and pangenomics to be inherently different. I would assume that metagenomics could be used both to define the pangenome (i.e., to find core and accessory genes in metagenomic assemblies) and to determine the abundances of subpopulations and/or specific genes or regions of the pangenome (through read mapping to metagenomic assemblies, SAGs, isolate genomes, or any combination of the above). Which, if any, of these possibilities apply to this study is not clear.

We agree with the reviewer that metagenomics alone can be useful to discuss pangenomes in the environment. However, the efficiency of this strategy is rather limited: The main distinction between the two endeavors is that pangenomes provide a framework to characterize shared gene content between closely related genomes with very high accuracy. In comparison, while metagenomic assembly and binning strategies are excellent tools to recover population genomes directly from the environment, they often fail to recover accessory genes and hypervariable regions of sub-populations. For instance, defining the Prochlorococcus pangenome through metagenomics alone would have been impossible given the current read lengths from state-of-the-art metagenomic surveys such as the TARA Oceans Project. That's why the two recent studies by Tully et al. (doi: 10.1101/162503) and Delmont et al. (doi: 10.1101/129791) which generated 2,991 population genomes from the TARA Oceans Project metagenomes only had 5 genome bins that resolved to Prochlorococcus despite the tremendous abundance of this taxon in the same metagenomic data as our study demonstrates. The same limitation will be true for other clades that maintain large populations with remarkable complexity in marine environments. Given the limitations of short reads, most accessory genes fail to assemble into large contigs, and the state-of-the-art binning algorithms fail to place the ones that are assembled into appropriate genome bins due to differences in their coverages compared to the rest of the population genome. That is why main points we were able to make in our study required a comprehensive focus on both pangenomes and metagenomes in a complementary manner.

-Metapangenomics is not defined, and I do not find it to be a helpful term. Consider removing it from the manuscript, including the title. Based on my reading of the abstract alone, it looks like metapangenomics is meant to describe an abundance-informed pangenome, and if so, why not call it something like that, with some useful meaning in the term itself?

We thank the reviewer for their input. To address their point, we removed the Results section that did not contribute to the introduction of this concept, and we included a more comprehensive definition of 'metapangenome' in the methods section. On the other hand, we respectfully disagree with their comment on the helpfulness of the term. Our writing experience in other studies (in which we use metapangenomes) convinces us that having a specific term results in a better flow as it did in the current study. The term 'metapangenome' may or may not be adopted by others, and indeed some may use 'abundance-informed pangenome' as their choice of wording.

Main Text:

-ln 32: complementary

The embarrassing typo is now fixed. We thank the reviewer.

[Figure]

A. Murat Eren, Ph.D.
ASSISTANT PROFESSOR, DEPARTMENT OF MEDICINE

**Knapp Center for Biomedical Discovery**
900 E. 57th Street, Mailbox 9, Room 9118, Chicago, IL 60637
P: +1-773-702-5935 / F: +1-773-702-2281 / meren@uchicago.edu

-ln 38-42: What does this sentence mean? Consider splitting it into two sentences. How does a metapangenome correlate with something? What is it correlating with? I am not sure that "traits" is an appropriate word – are the authors describing core and accessory genes here? What are "sub-clade demarcations"? How would these results differ from phylogenetic analyses (wouldn't phylogenetics by definition separate sub-clades?)? Do the authors mean some specific phylogenetic analyses that are typically performed at a coarser resolution? If so, please provide more context on the phylogenetic analyses.

Suggesting that a metapangenome can correlate with something was misleading, and we changed the sentence accordingly. We also split the sentence in two and better introduced the term of phylogenetics, as suggested by the reviewer. It now reads:

"*The resulting Prochlorococcus metapangenome revealed remarkable differential abundance patterns between very closely related isolates that belonged to the same phylogenetic cluster and that differed by only a small number of gene clusters in the pangenome. While the relationships between these genomes based on gene clusters correlated with their environmental distribution patterns, phylogenetic analyses using marker genes or concatenated single-copy core genes did not recapitulate these patterns.*"

Finally, we agree with the reviewer that "pangenomic traits" is not appropriate and we have replaced the term with "genes" or "gene clusters" depending on the context throughout the entire manuscript.

Main text:

-ln 54: "have been" should be "has been"

Fixed.

-Throughout: consider changing "shared" to "core" in the context of core genes across pangenomes, as this is more common in the literature and therefore more intuitive. If the authors mean shared among some but not all populations, then that should be explicitly mentioned, but I assume that they mean core genes shared among all populations.

Done. We now use the term of "core" throughout the text.

-ln 74-76: What does this mean? Wouldn't metagenomic assembly + read mapping do that too? It might help if you explain the particular utility of isolate sequences here.

We agree with the reviewer, and have removed "isolate" from this sentence and the following one. It now reads:

"*Metagenomic data also provide a means to quantify the abundance and relative distribution of genomes in environmental samples through read recruitment (Tyson et al., 2004; Dutilh et al., 2014; Eren et al., 2015). Although the environmental signal resulting from such analyses provides insights into the ecological niche of individual populations (Sharon et al., 2013; Bendall et al., 2016; Anderson et al., 2017; Quince et al., 2017), this approach alone does not reveal to what extent genes that may be linked to the ecology and fitness of microbes are conserved within a phylogenetic clade.*"

[Figure]

A. Murat Eren, Ph.D.
ASSISTANT PROFESSOR, DEPARTMENT OF MEDICINE

**Knapp Center for Biomedical Discovery**
900 E. 57th Street, Mailbox 9, Room 9118, Chicago, IL 60637
P: +1-773-702-5935 / F: +1-773-702-2281 / meren@uchicago.edu

-ln 77-78 (or earlier): Please define functional traits in the context of a microbial genome or pangenome. I think that you also mean isolate genomes here, so please change the end to "… mapping of closely related isolate genomes."

We removed "functional traits" to streamline the sentence. Also, because we now have removed "isolate" in previous sentences (see comment above), it became unnecessary to use this term here.

-ln 80-83: This sentence is a bit long and confusing and can probably be broken down. The authors can rework it for clarification as they see fit, but here are some examples of what I find confusing: What are "well-established practices in pangenomics"? Please give a few examples. What are "emerging opportunities from metagenomic data"? Is this just using metagenomic read mapping for abundance data? If so, just say that. What is a genome-centric framework (I assume that this involves the use of closed, isolate genomes), and how does that differ from what you would get from a combination of metagenomic assembly and binning to identify populations + read mapping across a number of metagenomes to get abundance estimates? What are "pangenomic traits" and how do you define which ones are "key"? Are "key" traits just those that are linked to "niche partitioning" and "population fitness", and if so, how do you determine that?

We thank the reviewer for their push for simplicity and clarity. We believe the section in question is now improved. The paragraph below is its final version, and is followed by our point-by-point responses to the questions reviewer asked:

> "*Combining well-established practices from pangenomics (identifying gene clusters and inferring relationships between genomes based on shared genes), with the emerging opportunities from metagenomics (the ability to track populations precisely across environments through genome-wide read recruitment) could provide a framework to investigate the ecological role of gene clusters that may be linked to the niche partitioning and fitness of microbial populations. To explore the potential of this concept, we developed a novel workflow within an existing open-source software platform (Eren et al., 2015), and characterized the metapangenome of Prochlorococcus isolates and single-cell genomes on a large scale*"

#What are "well-established practices in pangenomics"?

Identifying gene clusters and inferring relationships between genomes based on shared genes. We now have clarified this in the sentence.

#What are "emerging opportunities from metagenomic data"? Is this just using metagenomic read mapping for abundance data?

Correct. It is genome-wide read recruitments. This has been added.

#What is a genome-centric framework (I assume that this involves the use of closed, isolate genomes), and how does that differ from what you would get from a combination of metagenomic assembly and binning to identify populations + read mapping across a number of metagenomes to get abundance estimates?

[Figure]

A. Murat Eren, Ph.D.

ASSISTANT PROFESSOR, DEPARTMENT OF MEDICINE

**Knapp Center for Biomedical Discovery**
900 E. 57th Street, Mailbox 9, Room 9118, Chicago, IL 60637
P: +1-773-702-5935 / F: +1-773-702-2281 / meren@uchicago.edu

We realized that the term of "genome-centric" was confusing and decided to remove it. The workflow discussed in this context differs from the workflow precisely because the suggested strategy alone does not identify core and accessory genes between closely related genomes.

> #What are "pangenomic traits" and how do you define which ones are "key"? Are "key" traits just those that are linked to "niche partitioning" and "population fitness", and if so, how do you determine that?

Pangenomic traits was a confusing way of referring to genes or gene clusters in the pangenome. In agreement with the reviewer, we replaced every occurrence of 'pangenomic traits' with genes or gene clusters depending on the context throughout the text.

In addition, we acknowledge that the term of "key gene clusters" can be subjective and will depend on the biological questions asked. For instance, we consider that gene clusters correlating with environmental variables of interest, or connecting sub-clades with a unique fitness can be defined as key. In the present study, we have identified key gene clusters by comparing results from the pangenomic analysis with results from the metagenomic analysis (for the "niche partitioning" and "population fitness" gene clusters) as well as phylogenomic analyses (for the "clade-specific" gene clusters). Examples are described in the "Results" section. Nevertheless, to avoid any possible confusion, we have removed the term "key" from the sentence.

> #ln 85: please state exactly what you mean by integrating pangenomic and metagenomic data. Again, what is "pangenomic data" and what is "metagenomic data"? Are there better terms for these types of data in this context, for example, "… integrating population pangenomes from multiple isolate genome sequences with their abundance profiles across environmental samples from metagenomic read mapping?" That might not even be correct, but the point is that I do not understand.

We have changed the sentence accordingly to comments made by the reviewer. Especially, we now better introduce what pangenomics and metagenomics can provide.

-I think that the focus of both the abstract and introduction and maybe even the title should be on the need for and development of this software pipeline, as that seems to be the key novel result of the study, tested on Prochlorococcus as an example, right? [[later edit: wait, but the tool is Anvi'o, which is fabulous but not new; please use the introduction to very clearly walk the reader through what is known vs. what is new in this study, both in terms of the visualization software pipeline and the Prochlorococcus biology]]. It seems dangerous to imply that metagenomics has never been used to identify the ecological niches of specific subpopulations (for example, the Banfield lab has worked in that general area, at least in AMD systems; how would isolate pangenomes add further information in that context?) and much safer to say that your software and visualization pipeline can help to identify and show these differences more clearly.

Our study contains two main novelties. First, it introduces novel programs within anvi'o to perform pangenomic analyses as well as metapangenomic analyses. In addition, it provides a framework around an ecological question to discuss the complementary nature of these two strategies that have not been discussed in the literature in great detail before. Second, it describes novel ecological insights into *Prochlorococcus*, demonstrating the relevance of the overall strategy.

[Figure]

A. Murat Eren, Ph.D.

ASSISTANT PROFESSOR, DEPARTMENT OF MEDICINE

**Knapp Center for Biomedical Discovery**
900 E. 57th Street, Mailbox 9, Room 9118, Chicago, IL  60637
P: +1-773-702-5935 / F: +1-773-702-2281 / meren@uchicago.edu

To expand on the first point, our abstract now clarifies that our study "present[s] an integrated analysis and visualization strategy that provides an interactive and reproducible framework to generate pangenomes and to study them in conjunction with metagenomes". This novel functionality is implemented in the platform anvi'o in the form of new modules, but we do not think this is an important point to make in the abstract, or the user needs to know that it is in anvi'o if they do not intend to use it. Furthermore, we find it concerning that we may need to give up best software design principles to avoid criticism due to the fact that anvi'o itself is not new. We could have followed the common practice, and implement this package by creating a new codebase with a new name. It would have taken much less effort since it wouldn't have required much technical attention to observe best design principles. However, the approach of creating a new tool for every problem is a practice that dramatically impacts the user experience and convenience, and does not enable us interrogate complex data that require holistic approaches rather than independent analyses, which is one of the bottlenecks of our field. We believe 'anvi'o is not new' should not constitute a valid argument against the novelty of these new modules that provide novel opportunities to researchers. That said, we updated our conclusions statement to clarify the fact that "[in this study] we developed novel software solutions and analytical tools within the open-source software platform anvi'o to create and study metapangenomes with interactive visualization and inspection capabilities".

To expand on the second point, we agree with the reviewer that metagenomics has certainly been used to identify the ecological niche of subpopulations. We have now modified the sentence to clarify the point it intended to make, and included four relevant citations:

> *"Although the environmental signal resulting from such analyses provides insights into the ecological niche of individual populations (Sharon et al., 2013; Bendall et al., 2016; Anderson et al., 2017; Quince et al., 2017), this approach alone does not reveal to what extent genes that may be linked to the ecology and fitness of microbes are conserved within a phylogenetic clade."*

-ln 93: How many genomes? That number seems important if all of these genomes are going into your downstream analyses.

The article we cited describes the sequencing of 27 novel *Prochlorococcus* genomes using cultivation techniques. In addition to those, we included other genomes that were available on the NCBI at the time we started our study. Nevertheless, the significance of the referenced work is to introduce the five phylogenetic clades rather than a regularly growing number of genomes. We slightly modified the sentence to clarify this point by moving the citation to the very end of it.

-ln 95: Were these 16S rRNA gene amplicon surveys or otherwise not metagenomic studies? That seems like a worthwhile point of clarification to help make the case for the current study that links isolate genomes to metagenomic data.

They include different types of methodology: ITS amplicons, qPCR, and *in situ* hybridization using 16S rRNA-targeted oligonucleotides. Introducing all of these techniques would not contribute to the clarity and main objectives of the introduction. In addition, our study is not the first one to link genomes (including within *Prochlorococcus*) to metagenomic data, or to perform pangenomic analyses of *Prochlorococcus* isolates. We referred the reader to relevant studies in the "Introduction" and "Results" sections. That said, we have improved the end of the paragraph to clarify the contribution of our study. It now reads:

[Figure]

A. Murat Eren, Ph.D.
ASSISTANT PROFESSOR, DEPARTMENT OF MEDICINE

**Knapp Center for Biomedical Discovery**
900 E. 57th Street, Mailbox 9, Room 9118, Chicago, IL  60637
P: +1-773-702-5935 / F: +1-773-702-2281 / meren@uchicago.edu

> *"Yet, to the best of our knowledge, pangenomes have never been linked to metagenomes at an appropriate resolution to monitor the distribution of individual gene clusters. Monitoring individual gene clusters is essential to scrutinize their prevalence across multiple microbial genomes, and infer associations regarding their potential role in fitness and niche partitioning of microbial populations to which they belong."*

-ln 96: dynamics plural

Done.

-ln 97-98: Correlations between the "genomic traits" of isolates … are these just groups of genes that correlate with environmental variables? Were there correlations to variables other than HL and LL? If so, maybe call these "other environmental variables."

Authors of these studies took into consideration all environmental variables, including those that resulted in HL and LL demarcations, to study differentially occurring genomic traits. We believe calling those 'other environmental variables' could create confusion and may be misleading.

-ln 98-104: This is a long sentence, and I got lost halfway through. Do "these two groups" refer to the core and accessory genes? How many metagenomes? What does "independently" mean here, and what is "their differential occurrence"? Does "in Prochlorococcus populations" mean in the same 12 as at the beginning of the sentence? If so, change to "in the 12 Prochlorocuccus populations," otherwise define these populations. I don't understand the last part of the sentence at all. aybe summarize these three studies in three separate sentences and explain which part(s) of each are being included in the current analyses, and then explain clearly how the current study will expand on what is already known from these previous studies.

We have removed one citation for clarity, modified the wording, and have split the sentence into two. It now reads:

> *"A previous study by Coleman & Chisholm (2010) used a pangenome of 12 Prochlorococcus isolates to discuss the differential occurrence in Prochlorococcus populations between two sampling stations after identifying core versus accessory genes and observing that only a few genes differed significantly in abundance between the sites. In addition, Kent et al. (2016) showed a strong association between the Prochlorococcus accessory gene functions and the community composition of this lineage on a large scale using metagenomes from the Global Ocean Sampling expedition."*

Finally, to explain how our study does "*expand on what is already know*", we added the following sentence, which addresses the next point made by the reviewer below:

> *"Yet, to the best of our knowledge, pangenomes have never been linked to metagenomes at an appropriate resolution to monitor the distribution of individual gene clusters. Monitoring individual gene clusters is essential to scrutinize their prevalence across multiple microbial genomes, and infer associations regarding their potential role in fitness and niche partitioning of microbial populations to which they belong."*

[Figure]

A. Murat Eren, Ph.D.

ASSISTANT PROFESSOR, DEPARTMENT OF MEDICINE

**Knapp Center for Biomedical Discovery**
900 E. 57th Street, Mailbox 9, Room 9118, Chicago, IL  60637
P: +1-773-702-5935 / F: +1-773-702-2281 / meren@uchicago.edu

-ln 104-105: This seems like an important distinction. To this point, the general implication is that this is the first time that anybody has thought of exploring niche partitioning in pangenomes or metagenomes, yet here the authors say that the difference is that previous studies have not had resolution at the level of protein clusters. Again, please dispense with the fancy sentences and terms and use the introduction to tell the reader what has been done in the past that is relevant, both in terms of biology and visualization/software, and then explicitly state what knowledge gap will be filled by this study. For example, it would be useful to explicitly say why monitoring protein clusters is useful.

We agree with the reviewer that introducing what knowledge gap has been filled by this study is important. The sentence we added to respond to their previous concern, addresses this one, as well.

-ln 106: Again, what are "pangenomic traits"? Maybe I am just stuck on "traits" as an ecological term and the authors just mean similarities and differences across populations?

We have removed the term of "pangenomic traits" throughout the entire manuscript.

-ln 106: How do these 31 Prochlorococcus isolates relate to the 12 (or more?) populations described from previous studies above?

They correspond to the 12 genomes plus additional genomes that have been made publically available since then. They are not introducing new clades, but increasing the number of genomes available for each clades that were described before.

-ln 108: Please give an exact number for billions

Added. It was 30.9 billion reads.

-ln 110: Define ecological niche; is this just HL vs. LL here?

Correct. It now reads:

> "Our investigation revealed that closely related Prochlorococcus populations sharing the same high-light niche (i.e., near the surface) exhibit considerable differences in their relative abundance that could be explained by a small number of differentially occurring gene clusters."

-ln 109-115: These are results that do not belong in the Introduction. Consider either reworking to frame these as hypotheses (or similar) that will be explored in this study, or remove this.

Presence or absence of brief summaries in the last paragraph of the introduction seems to vary across published studies. Here are some random examples of peer-reviewed articles appeared in journals this week with summary statements at the end of their introduction: doi:10.1038/s41564-017-0065-7 (Nature Microbiology), doi:10.1038/ismej.2017.181 (ISMEJ), and 10.1186/s40168-017-0374-3 (Microbiome). We believe there is value to include a short summary of the main observations at the end of Introduction to communicate the context of the study to the reader early on. That said, we shortened our summary to minimize redundancy between the "Introduction" and "Results" sections, which now reads:

[Figure]

A. Murat Eren, Ph.D.
ASSISTANT PROFESSOR, DEPARTMENT OF MEDICINE

**Knapp Center for Biomedical Discovery**
900 E. 57th Street, Mailbox 9, Room 9118, Chicago, IL 60637
P: +1-773-702-5935 / F: +1-773-702-2281 / meren@uchicago.edu

> *"Our investigation revealed that closely related Prochlorococcus populations sharing the same high-light niche (i.e., near the surface) exhibit considerable differences in their relative abundance that could be explained by a small number of differentially occurring gene clusters. Finally, we extended our analysis of 31 isolates with 74 single-amplified genomes (SAGs) and revealed intriguing patterns within Prochlorococcus hypervariable genomic islands by quantifying the link between individual gene clusters and the environment."*

-ln 113 and 117: These are the first mentions of SAGs. This seems to be of abstract-level importance in how you are defining your pangenomes (i.e., a combination of SAGs and isolates). Or am I not understanding how your pangenomes were defined? After reading more of the results, I do not see much in the way of SAGs there, so how did you decide when to use isolates and when to use SAGs in your analyses? SAGs are presumably less complete genomes, so if a particular gene is not detected in a SAG, it does not necessarily mean that it is actually absent. The authors know this, I am sure, but if this is part of the rationale for using only isolates for some of the analyses, it should be mentioned.

We used SAGs only to investigate whether gene clusters that were core to all isolates but did not recruit reads from the environment could be attributed to cultivation bias:

> *"To investigate whether this could be due to a cultivation bias that selects for members from these populations with a certain set of sugar utilization genes, we analyzed 74 single amplified genomes (SAGs) from a study by Kashtan et al. (2014) (Supplementary Table 4). Our analysis revealed that these gene clusters also occurred in a large number of SAGs (75.7% to 81.1%) (Supplementary Table 4). Most interestingly, metapangenomic analysis of SAGs using the same metagenomic dataset and bioinformatics workflow we used for the isolates also revealed that all genes in these gene clusters were EAGs (Supplementary Table 4), consistent with our observations in the HL isolates, and ruling out the 'cultivation bias' hypothesis."*

They were not used to determine the "main" *Prochlorococcus* pangenome, largely due their lack of 'completion' also the reviewer pointed out.

-I think that Anvi'o can also be called out specifically in the Abstract and/or Introduction. It is not clear to me what aspects of the Anvi'o workflow are new in this study, though the Introduction suggests that this is a novel pipeline. Based on the text to this point, I was expecting the presentation of a novel workflow, and this needs to be made more clear. I think that a paragraph in the Intro with Anvi'o background would be appropriate – how has Anvi'o been used in the past, and what specifically is the new application here? It seems like more than just plugging new data into the software, so maybe a flowchart figure would help? There is a section of the methods dedicated to this, which is good, but I wonder if at least some of that should be moved to the main text, given that the pipeline is one of the key outputs of the study and not just an ancillary method.

We thank the reviewer for their input. The metapangenomic workflow is a novel addition to anvi'o, it can be used without using any other previously described concepts in anvi'o (such as metagenomic binning and refinement, or single-nucleotide variant analyses), and it offers a functionality that did not exist in anvi'o prior to this work. Besides the dedicated methods section the reviewer positively mentioned, we have now added a sentence in our conclusion statement to clarify that in this study "we developed novel software solutions and analytical

[Figure]

A. Murat Eren, Ph.D.

ASSISTANT PROFESSOR, DEPARTMENT OF MEDICINE

**Knapp Center for Biomedical Discovery**
900 E. 57ᵗʰ Street, Mailbox 9, Room 9118, Chicago, IL 60637
P: +1-773-702-5935 / F: +1-773-702-2281 / meren@uchicago.edu

tools within the open-source software platform anvi'o to create and study metapangenomes with interactive visualization and inspection capabilities". Besides this, anvi'o simply provides a framework for metapangenomics. Perhaps this is not the best analogy, but we would like to suggest that in a sense this is similar to how R provides a framework for DADA2 (doi:10.1038/nmeth.3869), which is an algorithm implemented within R. The same reason why DADA2 does not introduce how R has been used for other tasks before applies to anvi'o in this case. Clarifying how anvi'o has been used in the past will unlikely contribute to a better understanding of the concept our study introduces and discusses.

Besides this, our stance also has a more philosophical dimension. In our opinion, a focus on anvi'o and its previous applications could discourage members of the community from considering it merely as a framework to implement their own contributions in the future. In that vein, not putting an early focus on anvi'o diminishes the importance of our previous contribution, treats the platform the way R is treated in a sense, and invites the reader to focus on the concept rather than the platform itself. In summary, while there is no technical reason to include an anvi'o background to improve the reading experience, not doing so may have some positive impact.

-ln 134: Has phylogenomics not been done on these 31 genomes before?

We know that the internal transcribed spacer region has been used to determine the phylogeny of these genomes as it is mentioned in the "Results" section:

> "*Previous phylogenetic analyses using the internal transcribed spacer region (Biller et al., 2014b) placed LL genomes into polyphyletic clades (LL-I being an outlier), which was echoed by the phylogenomic analysis we performed in this study using 37 core genes (Figure 1).*"

We are not aware of any published study describing the relationships between a large set of *Prochlorococcus* genomes using phylogenomic strategies.

-ln 176-199: Is this the new part? If so, you could start with something along the lines of "The Anvi'o pangenomic workflow developed for this study consists of …"

We thank the reviewer for this suggestion. We modified the beginning of that section according to their suggestion.

-ln 179: What is a "genome of interest"? Is this just every genome that will be considered for a given analysis, i.e., 31 Prochlorococcus isolates for this study? [I see later that this is the case, so please rephrase to make this more clear]

This was indeed confusing. We replaced "of interest" with "under consideration".

-ln 234 and 264: What about the SAGs?

Good point. As we mentioned before we used SAGs for a very specific purpose. We have changed the headers of the two paragraphs the reviewer pointed out to minimize confusion. They now read "*Environmental distribution of Prochlorococcus isolate genomes*" and "*The pangenome of Prochlorococcus isolate genomes*".

[Figure]

A. Murat Eren, Ph.D.
ASSISTANT PROFESSOR, DEPARTMENT OF MEDICINE

**Knapp Center for Biomedical Discovery**
900 E. 57th Street, Mailbox 9, Room 9118, Chicago, IL 60637
P: +1-773-702-5935 / F: +1-773-702-2281 / meren@uchicago.edu

-ln 234-243: Has this been done already for Prochlorococcus in any of the TARA Oceans publications? I would guess so, but maybe not all 31 isolates were included. It would be worth clarifying what parts of this analysis are new vs. what just needed to be done again here to feed into the Anvi'o pipeline.

Even though this is a commonly used workflow, researchers often use different methodologies, such as the mapping of single copy core genes only, or the mapping of entire genomic databases that cover many lineages. So, while it is correct that we had to do this to feed mapping results into the anvi'o pipeline, to the best of our knowledge this particular analysis workflow (same set of genomes, same mapping strategy, same collection of metagenomes) has not been published elsewhere.

-ln 244-256: These specific clades have not been described anywhere. I realize that a description of each could get tedious, but is there something general that you could say along the lines of, e.g., "All LL lineages come from low-light niches and include subclades I-IV defined by x, y, z" Otherwise, the description of these clades is not particularly useful. The figures just say that these are "literature-defined" lineages, which is fine, but the authors could briefly elaborate on these clade distinctions in the text.

The main characteristics of the different clades are their adaptation level to low light and high light. We introduced this important aspect of the isolate genomes in the introduction. We have included the description of sub-clades to improve clarity. It now reads:

> "Cultivation efforts targeting *Prochlorococcus* resulted in the recovery of genomes that represent members from five major phylogenetic clades divided into groups that are adapted to high-light (sub-clades HL-I and HL-II) or low-light (sub-clades LL-I, LL-II, LL-III, and LL-IV) (Biller et al., 2014a)."

-ln 277-285: Cool!

It is always encouraging to see a reviewer excited. We thank the reviewer for their sincerity.

-ln 298-299: ECGs and EDGs -- do we really need more acronyms? I saw these again a couple of pages later and had to dig back to this section to remind myself of what they are. [and again when I came back to the manuscript after a break] When these acronyms appear again a couple pages later (ln 356), the next sentence has four different acronyms occurring eight times …

We agree with the reviewer that we certainly do not need more acronyms, however, "ECGs" and "EDGs" are critical concepts for the metapangenomic workflow. That said, we understand that acronyms can confuse the reader, and in an attempt to minimize the use of novel ones we did two things. First, we removed 'PCs' from the text, and replaced all occurrences of 'PCs' with 'gene clusters' by spelling it out without an acronym. In another attempt to simplify our language, we replaced the terms 'environmentally connected genes' (ECGs) and 'environmentally disconnected genes' (EDGs) with 'environmental core genes' (ECGs) and 'environmental accessory genes' (EAGs). We believe this is a more intuitive definition of these genes we discuss in our study, and remembering these acronyms will be much more easier for the reader. The only remaining acronyms in our study are "LL", "HL" and "HL+LL". We are happy with this outcome, and thank the reviewer for pushing us to improve the readability of the text.

[Figure]

A. Murat Eren, Ph.D.
ASSISTANT PROFESSOR, DEPARTMENT OF MEDICINE

**Knapp Center for Biomedical Discovery**
900 E. 57th Street, Mailbox 9, Room 9118, Chicago, IL 60637
P: +1-773-702-5935 / F: +1-773-702-2281 / meren@uchicago.edu

-ln 313: Okay, metapangenomics is finally defined! I still don't fully understand the utility of this term. Maybe it is just me, but I do not find the introduction of new terms and acronyms in nearly every new manuscript in this field to be helpful.

We do not dismiss the reviewer's sentiment as we often feel similarly. That said, we believe useless terms or acronyms will not survive the selection and will not be used by others. Hence, there is no harm in proposing a new term if it likely improves the communication.

-ln 313-320: What is the result here? The result cannot just be the figure; there has to be some interpretation or guidance for what the reader should be seeing.

We thank the reviewer for making this point. We have re-organized this section in our revision. The definition of metagengenome is not introduced in the "Material and Methods" section, and results related to the Figure 3 are now described in the first paragraph of the section to improve readability. It now reads:

> "*A metapangenome provides access to the environmental detection of individual genes in gene clusters, along with the ecological niche boundaries of individual genomes. The Prochlorococcus metapangenome revealed differences within the members of the Clade HL-II with respect to their rate of detection in the environment (Figure 3; see the interactive version at the URL http://anvi-server.org/p/JNIBAB). Interestingly, the organization of genomes in HL-II based on gene clusters matched their detection gradient within their niche, with the least abundant and the most abundant genomes in the metagenomic data being at the two extremes of the cluster that described the Clade HL-II (Figure 3, Supplementary Table 2). (…)*"

-ln 354-356: This seems like an important contribution, and it is buried near the end of the Results.

Our attempt to gradually introduce the concept of metapangenomics pushed some of the key aspects of the study to the second half of the results section. We hope the readers of our study will be as careful as our reviewers, and will not ignore the second half of our findings.

-ln 366-367: Change to plural

Done.

-ln 370-371: Please rephrase this sentence for clarity. What is "they"?

We replaced "*they*" with "*these gene clusters*".

-ln 420: Is this really only a little effort?!

We removed the qualitative statement "with little effort" from the sentence. That said, besides the hardware needs, performing the metapangenomic analysis of *Prochlorococcus* using data from TARA Oceans was somewhat trivial. To demonstrate the actual amount of effort in absolute clarity, we have now included a reproducible workflow at the beginning of our methods section.

[Figure]

A. Murat Eren, Ph.D.

ASSISTANT PROFESSOR, DEPARTMENT OF MEDICINE

**Knapp Center for Biomedical Discovery**
900 E. 57th Street, Mailbox 9, Room 9118, Chicago, IL 60637
P: +1-773-702-5935 / F: +1-773-702-2281 / meren@uchicago.edu

-ln 464-466: Okay, the authors have confirmed the obvious application that I mentioned above, which is that this can also be applied to metagenome-assembled genomes. Why were those not considered here? I don't think that this is a hole-in-the-paper offense, but I am puzzled, as it seems like a relatively easy addition that would boost the size of the available pangenome significantly.

We thank the reviewer for their progressive outlook. Our previous attempt to publish this workflow was heavily criticized due to suspicions over the quality and biological significance of metagenome-assembled genomes. We thought eliminating such concerns by only focusing on high-quality genomes could help reviewers scrutinize more relevant aspects of our study.

Figures:

These are nice. If the authors insist on keeping the ECG and EDG acronyms, please define them in each figure legend.

Done. We defined each acronym in figure legends for Figure 3 and Figure 4.

Experimental design

See above (Basic reporting section) for a few specific comments related to better defining the research question and knowledge gap(s) filled by this study in the Introduction.

We thank the reviewer very much for their time and comments that allowed us to strengthen our study.